# Metastatic Tumor Cell-Specific FABP7 Promotes NSCLC Metastasis via Inhibiting β-Catenin Degradation

**DOI:** 10.3390/cells11050805

**Published:** 2022-02-25

**Authors:** Qiaorui Bai, Xia Yang, Quanfeng Li, Weizhong Chen, Han Tian, Rong Lian, Ximeng Liu, Shuang Wang, Yi Yang

**Affiliations:** 1Department of Basic Medicine, Zhongshan School of Medicine, Sun Yat-sen University, Guangzhou 510080, China; baiqr@mail2.sysu.edu.cn; 2Department of Microbiology, Zhongshan School of Medicine, Sun Yat-sen University, Guangzhou 510080, China; yangx285@mail2.sysu.edu.cn (X.Y.); liuxm37@mail2.sysu.edu.cn (X.L.); wangsh298@mail2.sysu.edu.cn (S.W.); 3Cancer Institute, School of Basic Medical Science, Southern Medical University, Guangzhou 510515, China; liquanfeng02@gmail.com; 4Department of Pharmacology, Zhongshan School of Medicine, Sun Yat-sen University, Guangzhou 510080, China; chenwzh63@mail2.sysu.edu.cn; 5Department of Biochemistry, Zhongshan School of Medicine, Sun Yat-sen University, Guangzhou 510080, China; tianhan@mail2.sysu.edu.cn; 6Key Laboratory of Tropical Disease Control, Ministry of Education, Sun Yat-sen University, Guangzhou 510080, China; lianrong@mail2.sysu.edu.cn

**Keywords:** NSCLC, metastasis, FABP7, β-catenin, single-cell RNA sequencing

## Abstract

Metastasis accounts for 90% of cancer-related deaths and represents a prominent malignant feature in non-small cell lung cancer (NSCLC), while tumor cell-specific mechanisms and molecules pivotal for the metastatic capacity remain unclear. By analyzing single-cell RNA sequencing data, we found that fatty acid binding protein 7 (FABP7) was specifically up-regulated in tumor cells of metastatic NSCLC patients and might be a prognostic indicator for poor survival. Experimental studies based on NSCLC cell lines showed that FABP7 promoted the metastatic competencies of NSCLC cells in vitro and in vivo. Mechanistically, we demonstrated that FABP7 was important to canonical Wnt signaling activation and competitively inhibited the interaction between β-catenin and components of its cytoplasmic degradation complex, thereby repressing the phosphorylation-dependent ubiquitination and degradation of β-catenin. Our present study identifies FABP7 as a metastatic tumor cell-specific pro-metastatic gene and uncovers a previously unknown regulatory mechanism underlying Wnt hyperactivation via FABP7-impaired cytoplasmic β-catenin degradation, implicating a novel molecule in regulating NSCLC metastasis.

## 1. Introduction

Metastasis accounts for a vast majority of cancer-associated deaths. Non-small cell lung cancer (NSCLC) is the major histopathological subtype of lung cancer with a high metastatic rate of approximately 50% at the time of initial diagnosis [1]. Despite advances in therapeutic strategies in recent decades, the clinical outcome of metastatic NSCLC remains poor. More than 57% of patients die within one year after diagnosis and only 2–13% of patients survive for five years or longer [2,3].

Metastasis is a multi-step process during which cancer cells disseminate from primary tumors to seed and form new malignant colonies in distant tissues [4]. Many attempts have been made to understand mechanisms underlying the metastatic process of tumor cells by next-generation sequencing, most of which are bulk-tumor based. Such data provide blended average output from bulk samples where tumor cells only account for 20–40% on average [5,6,7,8]. It is not until recent years that single-cell RNA sequencing (scRNA-seq) has been developed and a precise characterization of every type of cell inside the tumor mass can be revealed [6,7,9,10,11,12]. By using this technique, a high heterogeneity of either malignant or non-malignant cells in metastatic tumors has been demonstrated, distinct immune or non-immune cell subtypes accounting for metastasis have been identified, and gene signature-based functional indications have been made at the single-cell level in various types of cancers, including NSCLC [6,12,13]. In spite of the above progression, tumor cell-specific molecules and mechanisms regulating NSCLC metastasis remain unclear.

The canonical Wnt/β-catenin signaling plays a vital role in cell proliferation, morphological change, and tissue development and maintenance [14,15]. Activation of this pathway is triggered by the binding of secreted Wnt ligands to Frizzled/LRP transmembrane receptors, which recruits members of β-catenin degradation complex to the cell membrane, thus preventing the degradation of β-catenin. Consequently, β-catenin accumulates in the nucleus, binds T cell factor/lymphoid enhancer-binding factor proteins (TCF/LEF), and regulates the expressions of serial downstream target genes [14]. In NSCLC, Wnt/β-catenin signaling is closely related to the initiation, proliferation, invasion, metastasis, and chemoresistance of the malignancy [16,17]. Enhanced expression of β-catenin has been found in NSCLC cell lines [18], and hyperactive Wnt/β-catenin signaling promotes the infiltration and colonization of NSCLC cells in brain and bone metastases [17]. It is of note that major causes of aberrant activation of Wnt signaling found in some other cancer types, such as mutations in β-catenin or APC, are rare in most NSCLC cases [16,18]. Thus, genetic changes cannot fully explain the activation of the canonical Wnt pathway in NSCLC, and the molecular mechanism mediating the activation of this pathway remains to be further investigated.

Fatty acid binding protein 7 (FABP7) is a brain-specific cytoplasmic protein that is associated with neuronal and glial cell differentiation under physiological conditions [19,20,21]. Like other FABP family members (FABPs), FABP7 facilitates exogenous fatty acid uptake into cells as well as their intracellular trafficking, which is essential for lipid metabolism and its related biological activities under both physiological and pathological conditions [22]. In cancers, FABP7 up-regulation and its diverse oncogenic functions have been found in glioma, breast cancer, melanoma, renal cell carcinoma, and colon cancer [21,23,24,25,26,27,28,29,30]. Nevertheless, the role of FABP7 in NSCLC, a prominently high metastatic rate, remains unknown. In addition, mechanisms underlying the metastasis-related properties in this malignancy have not been examined.

By analyzing transcriptomic changes specifically in metastatic NSCLC tumor cells at single-cell resolution in this study, we found that FABP7 was up-regulated prominently and promoted metastasis-related traits of NSCLC cells. Mechanistically, FABP7 enhanced metastasis was associated with inhibition of the interaction between β-catenin and CK1α or GSK-3β, resulting in suppressed phosphorylation-dependent ubiquitin-proteasomal degradation of β-catenin, thus enhancing β-catenin accumulation and subsequent Wnt/β-catenin signaling activation. These data elucidate a previously unidentified regulatory mechanism underlying Wnt/β-catenin activation and suggest a pro-metastatic role of FABP7 in NSCLC metastasis.

## 2. Materials and Methods

### 2.1. Tumor Specimens from Patients

Tumor specimens of lung cancer patients diagnosed at the Sun Yat-Sen University Cancer Center from 2000 to 2004 were obtained as described in our previous reports [31,32]. For the use of these clinical materials for research purposes, prior patient consent and approval from the Institutional Research Ethics Committee were obtained.

### 2.2. Tumor Xenografts In Vivo

All animal experimental procedures were approved by the Institutional Animal Care and Use Committee of Sun Yat-sen University. Female BALB/c-nu mice (8–10 weeks of age, 18–20 g) were used to investigate the metastatic effects of FABP7. To test local invasion, tumor cells were injected subcutaneously into the flanks of mice (1 × 10^6^ cells suspended in 100 μL sterile PBS, A549-Vector (left flank) and A549-FABP7 (right flank), 4 of each). To investigate the distant lung dissemination of FABP7, 1 × 10^6^ A549-Vetor cells or A549-FABP7 cells (suspended in 100 μL sterile PBS) were injected into the lateral tail vein (*n* = 5 for each group), and metastases were monitored by bioluminescent imaging with Spectrum Living Image 4.0 software (Caliper Life Sciences, Hopkinton, MA, USA). On day 28, mice were anesthetized and sacrificed, and tumors were resected, sectioned, and histologically examined by picric acid or H&E staining. H&E images were captured using the AxioVision Rel. 4.6 computerized image analysis system (Carl Zeiss, Jena, Germany).

### 2.3. Cell Culture

NSCLC cell lines and 293FT human embryonic kidney cell line HEK293FT were obtained from the Cell Bank of Shanghai Institutes of Biological Sciences (Shanghai, China) or American Type Culture Collection (ATCC, Manassas, VA, USA) and maintained in DMEM medium (Invitrogen, Carlsbad, CA, USA) supplemented with 10% fetal bovine serum (HyClone, Logan, UT, USA). All cell lines were authenticated by short tandem repeat (STR) fingerprinting at the Laboratory of Forensic Medicine of Sun Yat-sen University (Guangzhou, China) and were confirmed to be mycoplasma-free. 

### 2.4. RNA Extraction, Reverse Transcription and Real-Time PCR

Total RNA was extracted using the TRIzol reagent (Invitrogen, Carlsbad, CA, USA) according to the manufacturer’s instruction. cDNA was synthesized from total RNA using the GoScript™ Reverse Transcription Mix (Promega, Madison, WI, USA) The SYBR green-based real-time PCR (qPCR) program was 95 °C, 30 s; (95 °C, 10 s; 58 °C, 30 s) × 40 cycles; 95 °C, 15 s; 60 °C, 5 s; 95 °C, 5 s; 4 °C, hold. All results were normalized to reference gene GAPDH and relative quantification was calculated by using the 2^−ΔΔCt^ method. Primer sequences were: LEF1-Sense: 5′-AGAACACCCCGATGACGGA-3′;LEF1-Antisense: 5′-GGCATCATTATGTACCCGGAAT-3′;MMP7-Sense: 5′-GAGTGAGCTACAGTGGGAACA-3′;MMP7-Antisense: 5′-CTATGACGCGGGAGTTTAACAT-3′;MMP9-Sense: 5′-TGTACCGCTATGGTTACACTCG-3′;MMP9-Antisense: 5′-GGCAGGGACAGTTGCTTCT-3′;MYC-Sense: 5′-GGCTCCTGGCAAAAGGTCA-3′;MYC-Antisense: 5′-CTGCGTAGTTGTGCTGATGT-3′;GAPDH-Sense: 5′-GGAGCGAGATCCCTCCAAAAT-3′;GAPDH-Antisense: 5′-GGCTGTTGTCATACTTCTCATGG-3′.

### 2.5. Plasmids, siRNAs and Transfection

Coding sequences of FABP7 or β-catenin proteins with tag (Flag or HA) were generated by PCR subcloning and inserted into retroviral transfer plasmid pQcxip-puro (Clontech, Palo Alto, CA, USA). To deplete expression, siRNA oligonucleotides of FABP7 (si1: 5′-GGAGACAAAGTGGTCATCA-3′; si2: 5′-ACACGGAGATTAGTTTCCA-3′) or β-catenin (si1: 5′-CGCATGGAAGAAATAGTTGAA-3′; si2: 5′-ATCTGTCTGCTCTAGTAATAA-3′) were purchased from Ribobio (Guangzhou, China) and shRNA sequences were cloned into the pSupper plasmid to generate pSupper-FABP7-sh1-neo and pSupper-FABP7-sh2-neo. Transfection of plasmids or oligonucleotides was performed using the Lipofectamine 3000 reagent (Invitrogen, Carlsbad, CA, USA) according to the manufacturer’s instruction.

### 2.6. Cell Invasion and Migration Assay

NSCLC cells (3 × 10^4^) were plated in the upper Transwell chamber (Corning Costar Corp, Cambridge, MA, USA) coated without or with Matrigel (BD Biosciences, San Jose, CA, USA) and incubated at 37 °C for 22 h, followed by removal of cells inside the upper chamber with cotton swabs. Cells migrating or invading through the chamber membrane were fixed with a mixture of methanol and acetic acid (3:1), stained with crystal violet, photographed, and quantified in 5 random fields.

### 2.7. Immunoprecipitation

Cell lysates were prepared from 3 × 10^7^ 293FT cells transfected with HA-tagged-β-catenin in an NP-40-containing lysis buffer supplemented with protease inhibitor cocktail (Roche, Basel, Switzerland), and then immunoprecipitated with HA affinity agarose (Sigma-Aldrich, Burlington, MA, USA) overnight at 4 °C. Beads containing affinity-bound proteins were washed 5 times with immunoprecipitation wash buffer (150 mM NaCl, 10 mM HEPES pH 7.4, 0.1% NP-40). The precipitated protein was denatured, separated on SDS-polyacrylamide gels, and detected by WB analysis.

### 2.8. Immunofluorescence Assay

Cells were seeded on the coverslips in 24-well plates and fixed with 4% paraformaldehyde. After 20 min, cells were permeabilized with phosphate-buffered saline (PBS) containing 0.2% Triton X-100 (PBS-T) for 5 min and then blocked with 1% bovine serum albumin in PBS-T for 10 min. Immunostaining was performed using primary antibodies, rabbit anti-FABP7 and mouse anti-β-catenin (Abcam, Cambridge, MA, USA), overnight at 4 °C. After PBS washing for 3 times, the slides were incubated with Alexa Fluor 555 donkey anti-rabbit IgG and Alexa Fluor 488 goat anti-mouse IgG (Beyotime, Beijing, China), respectively, at room temperature for 1 h. The slides were counterstained with DAPI (Beyotime, Beijing, China) and images were captured using the Zeiss microscope (Carl Zeiss, Jena, Germany).

### 2.9. Luciferase Reporter Assay

After seeding in triplicates in 24-well plates and allowed to settle for 24 h, 293FT cells were transfected with 200 ng wild-type (CCTTTGATC, TOPflash) or mutated (CCTTTGGCC, FOPflash) TCF/LEF DNA-binding sites firefly luciferase reporter (Upstate Biotechnology, Lake Placid, NY, USA) plus 5 ng pRL-TK renilla luciferase reporter plasmids (Promega, Madison, WI, USA). Dual-Luciferase reporter assays were performed 48 h later according to the manufacturer’s protocol for the Dual Luciferase Reporter Assay Kit (Promega, Madison, WI, USA). The TOPflash contains wild-type TCF/LEF DNA-binding sites to detect the transcriptional activity of Wnt/β-catenin signaling. The FOPflash contains mutated TCF/LEF binding sites to further eliminate interference of noise. Luciferase activity was calculated by the formula:  (TOP firefly luciferase activity/renila luciferase activity)/(FOP firefly luciferase activity/renila luciferase activity).

### 2.10. Preparation of Nuclear and Cytoplasmic Extracts

Nuclear and cytoplasmic fractions were extracted by NE-PER™ Nuclear and Cytoplasmic Extraction Reagents (Thermo Fisher, Waltham, MA, USA) according to the manufacturer’s instructions. The cell pellet was washed with PBS (pH 7.4) 2–3 times, transferred to a 1.5 mL microcentrifuge tube, centrifuged at 500× *g* for 2–3 min, and resuspended in 200 μL ice-cold CER I. Subsequently, the tube was vortexed vigorously and incubated on ice for 10 min. Then, incubated cells were supplemented with 11 μL ice-cold CER II, vortexed for 5 s, and centrifuged for 5 min at 16,000× *g*. While the supernatant (cytoplasmic fractions) was immediately transferred to a clean pre-chilled tube, the precipitate was resuspended in 100 μL ice-cold NER and vortexed for 60 s every 10 min, for a total of 40 min. Then, the supernatant (nuclear fractions) was centrifuged at ~16,000× *g* for 10 min and immediately transferred to a clean pre-chilled tube. All extracts were stored at −80 °C until use.

### 2.11. Western Blotting Analysis

Protein extracts were resolved through SDS-PAGE, detected using anti-FABP7, anti-beta-catenin, anti-CK1 alpha and anti-GSK3 beta (Abcam, Cambridge, MA, USA), anti-HA, anti-E-cadherin, anti-N-cadherin, and anti-vimentin (Cell Signaling, Danvers, MA, USA) antibodies, respectively. Blotted membranes were stripped and re-blotted with an anti-p84 rabbit monoclonal antibody (Sigma, St. Louis, MO, USA) or anti-GAPDH mouse monoclonal antibody (Abcam, Cambridge, MA, USA) as a loading control.

### 2.12. Wnt3a Activation

FABP7 knockdown cells were replaced with the medium containing Wnt3a (R&D Systems, Minneapolis, MN, USA; 50 ng/mL). After 6–8 h, cells were subjected to immunofluorescence assay, luciferase reporter assay, Transwell assay, or Western blotting analysis.

### 2.13. Immunohistochemistry Assays (IHC)

Paraffin-embedded NSCLC tissues were dissected into consecutive slices and analyzed using IHC assay as previously described [33] with antibodies against FABP7 (Abcam, ab32423, Cambridge, MA, USA 1:50), MMP7 (Abcam, ab4044, Cambridge, MA, USA 1:40), and MYC (HuaBio, 0912-2, Hangzhou, China, 1:200), respectively. The immunostaining degree of indicated proteins was evaluated and scored by two independent observers with both the proportions of positively stained tumor cells and the staining intensities. The proportions of positively stained tumor cells were scored as: 0 (no positive tumor cells), 1 (<5%), 2 (5–25%), 3 (25–50%), and 4 (>50%). The intensities of staining were determined as: 0 (no staining), 1 (weak staining = light yellow), 2 (moderate staining = yellow-brown), and 3 (strong staining = brown). The staining index (SI) was calculated as the function of staining intensity × percentage of positive tumor cells, resulting in scores of 0, 1, 2, 3, 4, 6, 8, 9, and 12. Cutoff values for high and low expression of target proteins were chosen based on a measurement of heterogeneity using the log-rank test with respect to overall survival. SI ≥ 4 indicates high expression and SI ≤ 3, low expression. Chi-squared (χ2) tests were used for contingency tables.

### 2.14. Colony Formation Assay

A total of 2000 cells were plated in 12-well plates and cultured for 7–12 days. Colonies were stained with 1.0% crystal violet after fixation with 10% formaldehyde for 5 min and counted.

### 2.15. Single-Cell Sequencing Analysis

Published 10× Genomics Single Cell 3′ v2 sequencing data E-MTAB-6149 and E-MTAB-6653 were obtained from the EMBL-EBI database. Two samples were removed due to outdated 10× v2 chemistry (patients No.1 and 2). Raw fastq data were mapped to GRCh38 genome by Cell Ranger (version 3.0.2). The quality control was performed by R package Seurat (version 3.0.2) [34]. Criteria used for filtering cells were 200 < nFeature_RNA < 5000, nCount_RNA > 200, and mitochondrial genes expression < 10%. Cell cycle regulation-related genes were examined to be uninfluential for clustering. Raw counts were regressed by the mitochondrial gene expression to improve clustering.

### 2.16. Identification of Cell Types

Firstly, clusters were identified as different cell types by acknowledged cell-defining markers. When cluster identity was unidentified or controversial, characteristics of the cluster were extracted by *findmarker* (a function of Seurat) and further confirmed by literature review, considering CellMarker database [35], and ToppGene Suite [36].

### 2.17. Inference of Malignant Cells

R package infercnv (version 1.4.0) [37] was used to identify tumor cells. Epithelial cells from adjacent samples were used as normal references. Abnormal copy number variations (CNVs) patterns of epithelial cells from tumor tissues were inferred and used to divide cells into 6 clusters by unsupervised hierarchical clustering. For each cluster, the inferCNV score is defined as the sum of cell average inferred CNVs. Clusters with apparent CNVs patterns and higher inferCNV scores were identified as tumor cells.

### 2.18. Differentially Expressed Genes Analyses and Functional Enrichments

There were 993 lung adenocarcinoma (LUAD) and lung squamous cell carcinoma (LUSC) bulk sequencing (bulk-seq) of patient samples obtained from the TCGA dataset (accessed on October 2020). Differentially expressed genes (DEGs) were analyzed by *findmarker*. Significant DEGs were defined as adjusted *p*-value < 0.05. GO and KEGG analyses were performed and visualized by clusterProfiler (version 3.16.1) [38]. Metastasis-related traits were searched by following keywords: adhesion/cadherin/migration/invasion/metastasis. Metastasis-related pathways were searched by: Notch/Hippo/Hedgehog/TGF/WNT. GSEA was performed using GSEA software (version 4.1.0) [39]. Samples were grouped by FABP7 positive or negative expressions.

### 2.19. Statistical Analysis

For all other statistical analyses, SPSS 20.0 (IBM, Armonk, NY, USA) statistical software was used. Comparisons between two groups were performed by the two-tailed Student’s t-test. Significance of Kaplan–Meier survival curves was assessed by a log-rank test or directly generated by KM-Plotter [40]. Correlations were analyzed by using Pearson’s correlation. Error bars represent mean ± SEM. derived from three independent experiments. *p*-values < 0.05 were considered statistically significant.

## 3. Results

### 3.1. ScRNA-Seq Reveals the Tumor Cell-Specific Up-Regulation of FABP7 in Metastatic NSCLC Patients

To explore the molecular mechanism underlying NSCLC metastasis, the published scRNA-seq data E-MTAB-6149 and E-MTAB-6653 were analyzed [8], and primary tumors from patients with or without metastasis (three of each) were compared. After quality control (Appendix A), we analyzed 74,162 cells using the Seurat pipeline [34] and identified nine major cell types (T, NK, mixed T&NK, B, myeloid, fibroblast, endothelial, epithelial, and red cells) (Figure 1A and Appendix A) by acknowledged cell-defining markers and extracted characteristics of clusters. Myeloid, NK, and T cells, which are the most abundant and heterogeneous populations, were further subclustered and analyzed respectively (Figure 1B and Appendix A). Plasma cells were identified from B cells (Appendix A), and tumor cells were further identified from cells with epithelial features by the inferCNV algorithm (Figure 1C). In tumor tissues, there were 23,215 cells from non-metastatic patients and 27,724 cells from metastatic ones. The summary of cell proportions is shown in Appendix A. Subsequently, differentially expressed genes (DEGs) of each cell subtype between patients with and without metastasis were analyzed, followed by GO and KEGG analyses (Appendix A), and results showed numerous functional alterations in the tumor microenvironment between metastatic and non-metastatic tumor tissues.

To uncover the unique molecular mechanism that endows tumor cells to metastasize, 5880 malignant cells were further clustered into 15 subclusters, where cells from metastatic patients displayed prominently different clusterings from those of non-metastatic patients (Figure 1D). Analyses on DEGs in malignant cells between NSCLC patients with and without metastasis showed that 128 genes were significantly up-regulated and 179 genes were notably down-regulated. Among all of these DEGs, FABP7 was the most remarkably up-regulated one with 19.7 times higher expression in metastatic tumor cells (Figure 1E). It is of particular note that FABP7 was specifically expressed in malignant cells from metastatic tumor tissues (Figure 1F), suggesting its potential role in promoting NSCLC metastasis. 

Most FABPs have similar sequences, structures, and fatty acid-binding functions [22]. Accordingly, we explored the expression of FABP7 and other FABPs in various cell types by analyzing scRNA-seq data and found that only FABP3-7 can be detected (Appendix A). By comparing tumor with adjacent tissue (Appendix A), or comparing a metastatic tumor with non-metastatic tumor (Figure 1G), we observed that FABP3 and 4 were expressed in a fraction of myeloid cells (Appendix A), and FABP4 displayed a marginal reduction in cells from metastatic tumor tissues (Figure 1G). FABP6 level was extremely low in most cell types (Appendix A), while FABP5 was widely expressed in various types of cells and was slightly increased in epithelial ones from metastatic NSCLC (Figure 1G). Notably, among these FABPs, FABP7 displayed the most unique and specific expressional pattern in metastatic tumor cells (Figure 1G and Appendix A), and showed the most prominent correlation between expression alteration and clinical prognosis (Figure 1G,H and Appendix A). The results indicate that FABP7 is a potential pro-metastatic FABP family member in NSCLC.

### 3.2. FABP7 Promotes Metastasis-Related Traits and Indicates a Worse NSCLC Outcome

To preliminarily assess whether FABP7 promotes metastasis, gene ontology (GO) analysis as well as gene-set enrichment analysis (GSEA) were performed based on the above scRNA-seq data. As shown in Figure 2A, DEGs between FABP7 positive and negative tumor cells displayed significantly overrepresented GO categories related to changes in metastatic competency (cell adhesion/migration/spreading), and GSEA suggested positive correlations between FABP7 and tumor metastasis (Figure 2B and Appendix A). Consistent results were also observed by using the TCGA NSCLC dataset (Appendix A), indicating that FABP7 is closely related to NSCLC metastasis. To experimentally validate its metastasis-facilitating ability, we constructed two NSCLC cell lines (A549 and H1975) with stable overexpression or knockdown of FABP7, followed by examination of expressional efficiencies (Figure 2C). We found that ectopic FABP7 decreased epithelial marker (E-cadherin) but increased mesenchymal markers (N-cadherin and Vimentin), while silencing FABP7 had the opposite effects (Figure 2D). Moreover, transwell assays showed that more NSCLC cells with ectopic expression of FABP7 passed through the cell compartment in the presence or absence of matrigel, when compared with their vector control. In contrast, NSCLC cells with depletion of endogenous FABP7 displayed the opposite behavior (Figure 2E). Interestingly, the expressional manipulation of FABP7 did not affect NSCLC growth using the colony assay (Appendix A). The above data suggest that both migrating and invading properties, but not growth rate, are facilitated by FABP7 in NSCLC cells. In mice, subcutaneously xenografted or tail vein-injected human NSCLC tumors cells confirmed FABP7-promoted local invasion (Figure 2F) or FABP7-facilitated distant metastases (Figure 2G,H), respectively. Furthermore, analyses on metastasis-free survival from datasets of MSKCC [41] showed that high levels of FABP7 were associated with shorter time to the first occurrence of NSCLC metastasis (Figure 2I). These results indicate that FABP7 promotes metastasis-related phenotypes and implicates poor prognoses of NSCLC. 

### 3.3. FABP7 Promotes and Mediates Activation of Wnt/β-Catenin Signaling in NSCLC

Next, we sought to investigate the molecular mechanism underlying the identified pro-metastatic effects of FABP7 at either single-cell or bulk-based level by analyzing both scRNA-seq and TCGA NSCLC datasets. Since FABP7 is known as a fatty acid transporter [22], we explored associations between FABP7 and lipid-related pathways in tumor cells but failed to find significant connections. qPCR validations on lipid droplets markers [42,43] have also implied this irrelevance (Appendix A). Interestingly, we found that the expression of FABP7 correlated remarkably with the activity of canonical Wnt signaling by GO and GSEA analyses (Figure 3A,B and Appendix A, and Appendix A), but not other metastasis-related pathways (Notch, Hippo, Hedgehog, and TGF-β). These data suggest a possibly unique regulatory mechanism of canonical Wnt pathway by FABP7. In line with this indication, experimental validation showed that both β-catenin transcriptional activities and target genes of Wnt/β-catenin pathway [44,45,46] were markedly enhanced or diminished, respectively, by ectopic expression or knockdown of FABP7 (Figure 3C,D). Moreover, FABP7 levels were closely associated with the expression of several well-known β-catenin target genes according to TCGA RNA-seq analyses (Figure 3E) and IHC staining on clinical specimens (Figure 3F). Furthermore, β-catenin localization was examined using subcellular fractionation assays and immunofluorescence staining, and we found that β-catenin level was increased in the nucleus when FABP7 was overexpressed (Figure 3G,H). These data suggest that FABP7 is able to activate Wnt/β-catenin signaling.

In addition to the above findings, we observed that silencing β-catenin, the central hub of canonical Wnt signaling, remarkably inhibited FABP7-promoted aggressiveness and FABP7-changed EMT markers (Figure 4A,B), indicating the importance of β-catenin in metastasis facilitated by FABP7. We also found that FABP7 depletion partially impaired both luciferase reporter activity and β-catenin nuclear accumulation increased by Wnt-3a, a ligand that activates the canonical Wnt pathway (Figure 4C,D). Upon Wnt-3a stimulation, NSCLC cells became more aggressive in migration and invasion with decreased epithelial but increased mesenchymal markers, while these effects were partially abrogated by FABP7 knockdown (Figure 4E,F), suggesting that FABP7 may be important, at least in part, to the Wnt-3a-activated β-catenin pathway. Altogether, these results indicate that FABP7 mediates activation of canonical Wnt pathway as a pivotal component of the Wnt/FABP7/β-catenin axis.

### 3.4. FABP7 Stabilizes β-Catenin by Inhibiting Its Ubiquitin-Proteasomal Degradation

β-catenin proteins were increased by FABP7 in not only the nucleus, but also the cytoplasm (Figure 3G). Therefore, we investigated whether FABP7 can up-regulate β-catenin and demonstrated the dramatic enhancement of β-catenin protein level by FABP7 in whole-cell lysate (Figure 5A). Since destruction complex-mediated ubiquitin-proteasomal degradation is pivotal for regulating cellular β-catenin [47,48], we wondered if FABP7 can affect β-catenin degradation and observed that FABP7 inhibit β-catenin turnover in the presence of protein synthesis inhibitor cycloheximide (CHX) (Figure 5B). β-catenin protein levels reduced by FABP7 depletion could be rescued by proteasomal inhibitor MG132 (Figure 5C). In comparison with the control group, silencing of FABP7 caused a remarkable improvement of polyubiquitination in the β-catenin immunoprecipitation (Figure 5D), indicating the ubiquitin-proteasomal dependent post-translational modulation of β-catenin by FABP7. Notably, endogenous FABP7 could be detected in β-catenin immunoprecipitation (Figure 5E), implicating their potential interaction. FABP7 remarkably inhibited the binding between β-catenin and components of the degradation complex, CK1α and GSK-3β, two Ser/Thr kinases responsible for β-catenin phosphorylation and mediating subsequent ubiquitination (Figure 5F). In addition, the ectopic expression of FABP7 reduced the phosphorylation of β-catenin (Figure 5G). These data indicate that FABP7 stabilizes β-catenin protein by obstructing its binding to CK1α and GSK-3β, thus inhibiting the subsequent ubiquitin-proteasomal degradation of β-catenin.

## 4. Discussion

Even at a very early stage, tumor cells in primary sites have acquired metastatic ability before distant metastasis [4]. Distinguishing these highly metastatic cells within a complex tumor mix and targeting them may be very important for preventing tumor metastasis. Activation of canonical Wnt pathway is well-known by bulk-based analyses [17,49,50]. In our present study, we demonstrate the metastatic malignant cell-specific Wnt/β-catenin activation at the single-cell level for the first time via scRNA-seq analysis, which enables the molecular distinction of malignant cells within a complex tumor mix. In addition, the single-cell-based analysis leads to the discovery of a tumor cell-specific and metastasis-specific expression pattern of FABP7. Interestingly, such a change of FABP7 expression cannot be found by bulk-based methods, e.g., in the TCGA dataset between metastatic versus non-metastatic NSCLC tumor masses (data not shown). This may be due to the high heterogeneity of the tumor mass and that bulk-based analyses only provide an average of diverse constituent cells, which may obscure crucial alternations.

FABPs have been reported to participate in fatty acid transportation to specific cellular compartments and to coordinate metabolic as well as inflammatory pathways [51]. Abnormal expression of FABPs occurs widely in distinct cancers, suggesting their role in tumor progression [22,51]. The scRNA-seq results show that only FABP3/4/5/7, and a very small amount of FABP6, are expressed in NSCLC. In this malignancy, FABP5 was reported to facilitate tumorigenesis and metastasis [52,53], while the functions of FABP3 and 4 are less clear and even become controversial [54], and the role of FABP6 and 7 remain unclarified in NSCLC. In fact, our work firstly reports the pro-metastatic function of FABP7 in NSCLC. Intriguingly, Umaru et al. have found that FABP7 can foster proliferation in melanoma [55], but FABP7 did not display the same effect in NSCLC (Appendix A), suggesting cancer context-dependent functions of FABP7. Moreover, Umaru et al. have reported the relationship between FABP7 and Wnt/β-catenin pathway [55]. It is the first study showing this connection but it does not clarify how FABP7 activates this signaling. In our present research, we demonstrate that FABP7, by competitively inhibiting β-catenin binding to kinase components of its degradation complex, stabilizes β-catenin protein and subsequently activates the signaling pathway. These findings elucidate, for the first time, the mechanism underlying FABP7-regulated Wnt/β-catenin signaling.

FABP7 distributes in both nucleus and cytoplasm [56]. Nuclear FABP7, which interacts with nuclear receptor PPAR-γ and is important for acetyl-CoA metabolism to modify histone structure in the nucleus [23,57], may contribute to metastatic traits of malignant glioma [23,56,58,59,60]. In our present study, the predominantly cytoplasmic distribution of FABP7 and its impairment on β-catenin degradation, which occurs in cytoplasm, indicate not only a cytoplasmic function of FABP7 in tumor metastasis, but also that the cytoplasmic levels of FABP7 may be important for predicting β-catenin stabilization, canonical Wnt signaling activation, aggressiveness, and prognosis of NSCLC. This notion is consistent with a large cohort of 1494 patients showing that breast cancers with cytoplasmic FABP7 expressions have shorter survival than those with nuclear expression [61]. Moreover, we clarify for the first time that β-catenin degradation can be hampered by FABP7, suggesting a previously unidentified mutation-independent mechanism of Wnt/β-catenin signaling deregulation in NSCLC. Since hyperactivation of this pathway in most NSCLC cases cannot be attributed to such genetic changes [62], our finding may provide a new potential prognostic strategy for NSCLC metastasis.

As a fatty acid transporter, FABP7 can promote lipid droplet formation by increasing fatty acid uptake under hypoxia in various tumors [21,63,64]. However, in NSCLC considered in our present study, we did not find the association between FABP7 and lipid droplet formation, evidenced by failure in functional enrichment and unaltered expression of lipid droplet related genes in NSCLC tumor cells between metastatic and non-metastatic tumors, or between FABP7 positive and negative tumors. These results suggest that the role of FABP7 on lipid formation may be cancer context-dependent. In fact, it has been reported that fatty acid uptake promoted by some other FABPs (1/2/4) relies on environment or cell type [65]. It is of note that the main ligand of FABP7 (DHA) has a controversial effect on lipid droplet formation [66,67]. Thus, further investigation on mechanisms underlying a tumor context-dependent role of FABP7 in formation of lipid droplets will be interesting and necessary.

In conclusion, FABP7 is up-regulated specifically in tumor cells of metastatic NSCLC and endows tumors with metastatic potency via impairing the ubiquitin-proteasomal degradation of β-catenin to activate canonical Wnt signaling. Our work identifies a tumor cell-specific enhancement of pro-metastatic FABP7 and elucidates a previously unidentified mechanism for Wnt/β-catenin signaling deregulation in NSCLC.

## Figures and Tables

**Figure 1 cells-11-00805-f001:**
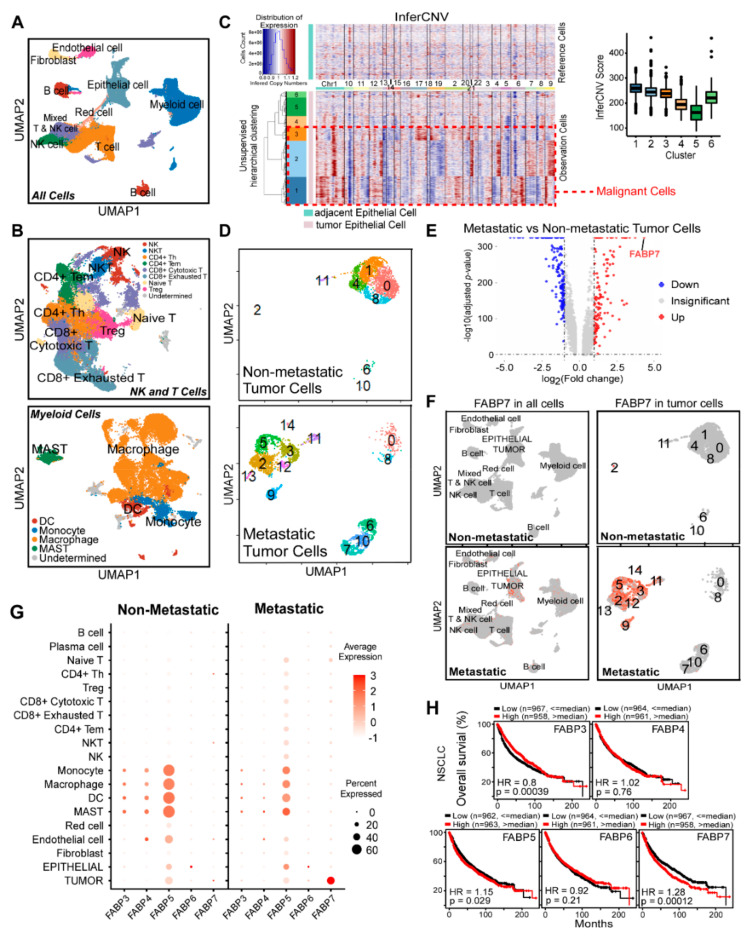
FABP7 is highly expressed in NSCLC cells. (**A**) UMAP visualization of all cells in the single-cell dataset colored by identified cell types. (**B**) UMAP visualization of NK and T cells (top) and myeloid cells (bottom) in the single-cell dataset colored by identified cell types. (**C**) Identification of tumor cells. Inferred copy number variations (CNVs) patterns of each chromosome (shown by different color bars) were compared between normal cells (Reference Cells) and epithelial cells from tumor tissues (Observation Cells). A higher inferCNV score (right) represents a higher CNVs degree of the cluster. The red dash line box shows malignant cells with obvious CNVs and higher inferCNV scores. (**D**) UMAP visualization of tumor cells from non-metastatic (top) or metastatic (bottom) patients. (**E**) Volcano plot of differentially expressed genes (DEGs) between tumor cells of metastatic and non-metastatic patients. Expressions with adjusted *p*-value < 0.05 are shown in red (log_2_ Fold change > 1) or blue (log_2_ Fold change < −1), while those with adjusted *p*-value > = 0.05 are shown in grey. (**F**) FABP7 Expression of all main cell types (left) or tumor cells (right) split by patients with or without metastasis. Expression levels of FABP7 are labeled by the intensity of red color. (**G**) Average expressions of FABP3-7 in indicated cell types split by patients without (left) or with (right) metastasis. (**H**) Kaplan-Meier overall survival (OS) analyses of FABP3-7 in NSCLC based on KM-plotter.

**Figure 2 cells-11-00805-f002:**
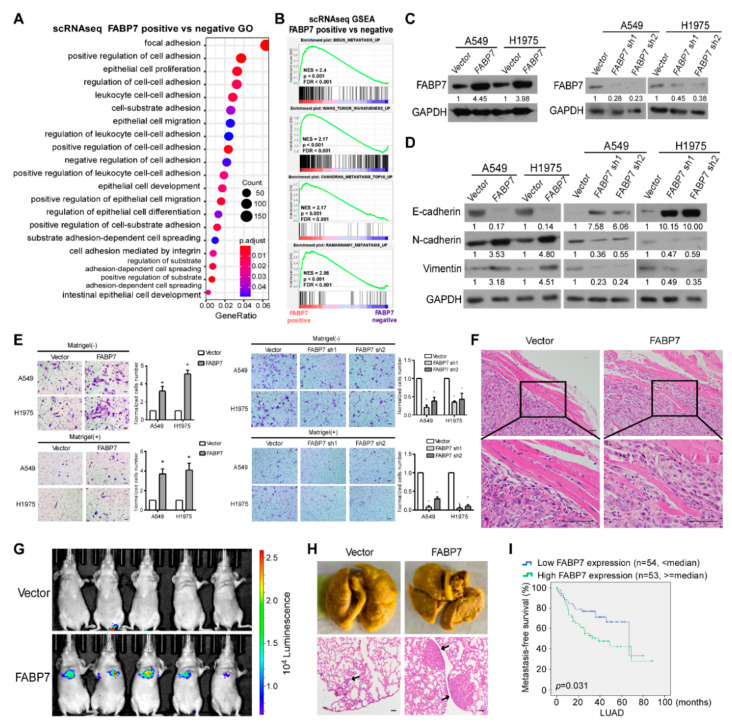
FABP7 contributes to NSCLC metastasis and indicates poor prognoses. (**A**) Metastasis-related biological functions enriched by Gene Ontology (GO) analysis. DEGs from scRNA-seq comparing FABP7-positive with FABP7-negative tumor cells were used for the analysis. (**B**) Gene Set Enrichment Analysis (GSEA) for DEGs between FABP7-positive and FABP7-negative malignant cells from scRNA-seq dataset. (**C**) Expression efficiencies of FABP7 stable over-expression (left) and knockdown (right) cell lines. Representative images from three independent experiments are shown. (**D**) Western blot analyses of EMT markers in NSCLC cells with stable over-expression or knockdown of FABP7. Representative images from three independent experiments are shown. (**E**) NSCLC cells were plated in chambers without or with matrigel respectively to perform transwell assays. Migratory (without matrigel) or invasive cells (with matrigel) were quantified in five random fields. Scale bar: 50 μm. Error bars represent the means ± SD derived from three independent experiments. Statistical analyses were performed by two-sided Mann-Whitney test (*: *p* < 0.05). (**F**) Representative H&E staining images (insets: high-magnification images) of xenografts caused by vector or FABP7 A549 cells. *n* = 4, Scale bar: 50 μm. (**G**,**H**) Indicated A549 cells (1 × 10^6^) were injected via the tail vein. Representative bioluminescent images (**G**), picric acid staining and H&E staining (**H**) of lung metastases are shown. *n* = 5, Scale bar: 50 μm. (**I**) Metastasis-free survival (MFS) Kaplan-Meier analyses of LUAD from the MSKCC dataset.

**Figure 3 cells-11-00805-f003:**
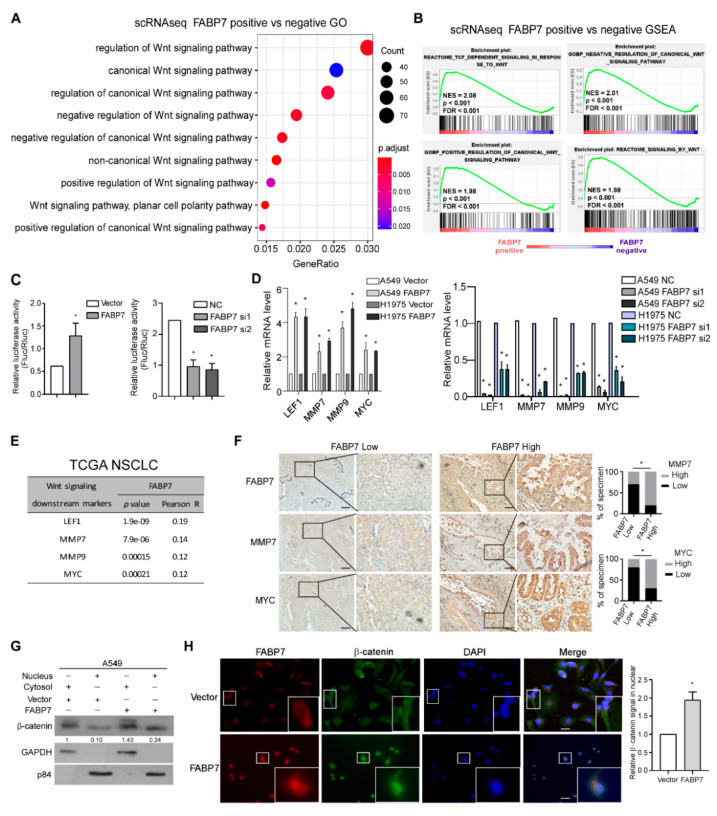
FABP7 transactivates Wnt/β-catenin signaling. (**A**) Wnt signaling pathway enriched by GO analysis. DEGs from scRNA-seq comparing between FABP7-positive and FABP7-negative tumor cells were analyzed. (**B**) GSEA for Wnt signaling pathway related gene sets from the scRNA-seq dataset, grouped by the FABP7 expression. (**C**) Relative luciferase assay examines effects on Wnt/β-catenin activation by over-expression (left) or silencing (right) of FABP7. Error bars represent the mean ± SD derived from three independent experiments. Statistical analyses were performed by two-sided Mann-Whitney test, *: *p* < 0.05. (**D**) Relative mRNA levels of Wnt/β-catenin downstream targets in FABP7 overexpressing (left) or silencing cells (right). Error bars represent the mean ± SD derived from three independent experiments. Statistical analyses were performed by two-sided Mann-Whitney test (*: *p* < 0.05). (**E**) Expression correlations between FABP7 and Wnt/β-catenin signaling targets in NSCLC cohort from TCGA dataset. (**F**) Representative images of immunohistochemistry consecutive slices staining (insets: high-magnification images) showing protein levels of FABP7 and Wnt/β-catenin signaling target genes in primary NSCLC tumors with low (*n* = 10) or high (*n* = 10) FABP7 expressions. Scale bar: 50 μm. Chi-square tests, *: *p* < 0.05. (**G**) Nucleo-plasma separation and western blotting assays show the changes of β-catenin expression at subcellular level by FABP7 ectopic expression in A549 cells. Representative images from three independent experiments are shown. (**H**) Representative IF images of three independent experiments (insets: high-magnification images) and quantitative results of alterations in β-catenin subcellular localization by FABP7 overexpression in A549 cells. Scale bar: 10 μm. Error bars represent the means ± SD derived from three independent experiments. Statistical analyses were performed by two-sided Mann-Whitney test (*: *p* < 0.05).

**Figure 4 cells-11-00805-f004:**
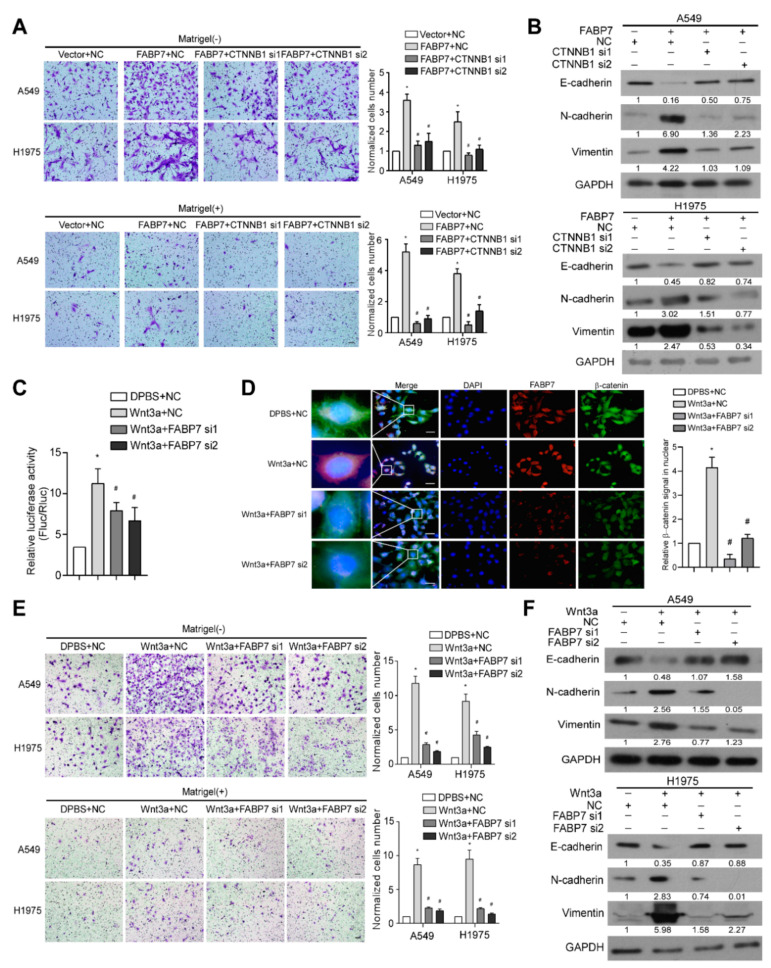
FABP7 is important for the activation of Wnt/β-catenin signaling. (**A**) Tanswell assays were performed in indicated NSCLC cells. Representative images from three independent experiments (left) are shown. Migratory or invasive cells were quantified in five random fields. Scale bar: 50 μm. *: *p* < 0.05 vs. vector + negative control (NC), #: *p* < 0.05 vs. FABP7+NC. (**B**) Western blot analyses of EMT markers in indicated NSCLC cells. Representative images from three independent experiments are shown. (**C**,**D**) Results of relative luciferase assays (**C**) and representative IF images (insets: high-magnification images) from three independent experiments (**D**) in H1975 cells. *: *p* < 0.05 vs. vehicle+NC, #: *p* < 0.05 vs. Wnt3a+NC. Scale bar: 10 μm. (**E**) Tanswell assays were performed in indicated NSCLC cells. Representative images are from five random fields of transwell assay. Migratory or invasive cells were quantified in five random fields. Scale bar: 50 μm. *: *p* < 0.05 vs. vehicle+NC, #: *p* < 0.05 vs. Wnt3a+NC. (**F**) Western blot analyses on EMT markers in indicated NSCLC cells. Representative images are shown from three independent experiments. All error bars represent the mean ± SD derived from three independent experiments, and statistical analyses were performed by two-sided Mann-Whitney test.

**Figure 5 cells-11-00805-f005:**
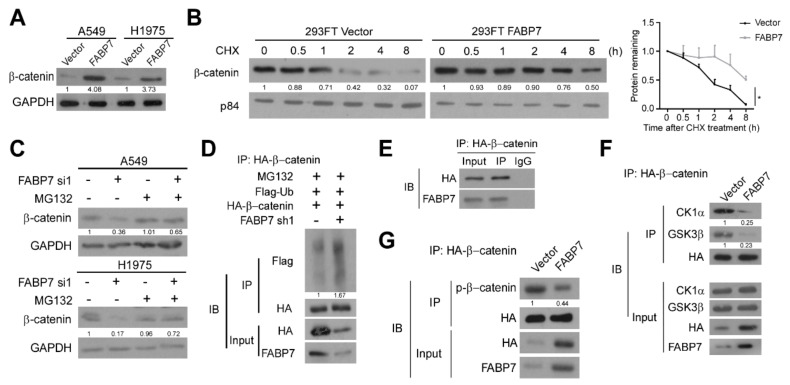
FABP7 stabilizes β-catenin by inhibiting its ubiquitin-proteasomal degradation. (**A**) Western blotting analyses on β-catenin levels in indicated NSCLC cells stably expressing vector or FABP7. (**B**) Western blotting analyses (left) and corresponding quantitation (right) on β-catenin stability in 293FT cells with indicated treatments in the presence of CHX. Error bars represent the means ± SD derived from three independent experiments, and statistical analyses were performed by two-sided Mann-Whitney test (*: *p* < 0.05). (**C**) Alteration of β-catenin stability after indicated treatments in the presence of MG132 with or without FABP7 depletion by Western blotting analyses. (**D**) Co-immunoprecipitation (co-IP) assays in stable FABP7 knockdown A549 cells show changes of β-catenin polyubiquitination in the presence of MG132. (**E**) Interaction between FABP7 and purified β-catenin protein in A549 cells. (**F**) Interaction between β-catenin and destruction complex components upon FABP7 overexpression in A549 cells. (**G**) Co-IP assays show changes in β-catenin phosphorylation by FABP7 overexpression in A549 cells. Representative images are from three independent experiments.

## Data Availability

Publicly available datasets were analyzed in this study. The scRNA-seq data can be found here: https://www.ebi.ac.uk/arrayexpress/experiments/E-MTAB-6149/ (accessed on June 2019) and https://www.ebi.ac.uk/arrayexpress/experiments/E-MTAB-6653/ (accessed on June 2019). The TCGA bulk-seq data of NSCLC patients can be found here: https://portal.gdc.cancer.gov/ (accessed on October 2020). The datasets used in this study are available from the corresponding author on reasonable requests.

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
