# Peer review of "Metastatic Tumor Cell-Specific FABP7 Promotes NSCLC Metastasis via Inhibiting β-Catenin Degradation"

_cells, 2022, doi:10.3390/cells11050805_

Round 1
Reviewer 1 Report
Qiaorui Bai et al. described the role of FABP7 in promoting metastasis via Wnt signaling in NSCLC. The mechanism, well known in melanoma models (ref 63) is first described in NSCLC. The paper is very interesting and the experiments were well performed. Hovewer, there are several points that need to be addressed.
- The results (especially for the in vitro experiments) should be described more in detail.
- Have you seen effects of FABP7 overexpression on cell proliferation? These data are missing and I suggest adding them.
- Since FABP7 is involved in lipid transport I suggest also evaluating also a role of lipid accumulation (such as lipid droplets formation) in promoting metastasis in NSCLC cells.
- Considering that the authors indicated FABP7 as a therapeutic targeting for the treatment of metastatic NSCLC, I suggest to evaluate the effects of pharmacological inhibition of FABP7 on cell proliferation, migration and invasion in NSCLC cells.
- The resuls should be better discussed in the Discussion section.
- The manuscript must be carefully corrected for typographical errors.
Reviewer 2 Report
In the present paper, the authors showed that FABP7 is able to modulate Wnt/ß-catenin pathway thus promoting metastasis in NSCLC.
I think that the most important weakness is that a connection between FABP7 and Wnt/ß-catenin signalling in regulating metastasis has been already exposed for example by Umaru et al. (ref 63 of this paper) in melanoma, especially for the in vitro section. So, I do not agree with the sentence in rows 77-81.
Introduction
Rows 57-60. I think that the mechanism of Wnt signaling is not so clear. I think that, after the Wnt-Frizzled interaction, the activated receptor recruits members of β-catenin degradation complex and not β-catenin itself. Please rewrite correctly.
Material and Methods
Probably due to limits in characters number, material and methods are not exhaustive. If additional space is allowed, I would suggest more details.
Results
- Rows 295-305. How did the authors explore the expression of other FABPs? In the same datasets used for sc-RNA seq analyses, where they found that ‘FABP7 was the most remarkably up-regulated gene’? I think they should use, if available, other datasets.
- Rows 313-314. On my opinion, indicating a pro-metastatic role of FABP7 just because they found a positive correlation is excessive. Please rephrase.
- For almost all the in vitro experiments, the authors used two NSCLC cell lines where they overexpressed or silenced FABP7. In parallel with this sort of ‘isogenic’ systems, could the authors not have used a panel of NSCLC cell lines with different ‘physiological’ expression of the protein?
- Rows 319-322. I think that tail vein-injected cells is not the right model to study distant metastasis: lung is the primary engrafting site of tail vein-injected cells.
- Figure 2C. I think that there is a too high difference in the FABP7 expression between the vector samples of the FABP7 overexpression and the silencing of the same cell line. I understand that if there is an overexpression the endogenous level could seem lower, but in these western blot the overexpressed band is not so high and the vector band is absent.
- Figure 2I. Where is the OS graph they mentioned in the text (rows 322-323)?
- Rows 353-354. Please add some references. about Wnt/β-catenin target genes.
- Figure 3c.What about the luciferase activity performed with the mutated TCF/LEF binding sequence vector mentioned in Material and Methods section (rows 167-173)? In which cells were the experiments performed?
- Figure 2D, right panel. Is this a relative quantification? Which is the reference sample? Could the controls not have been all equal to one?
- Figure 2E. What about H1975?
- Figure 3H. Could the authors use consecutive slices to perform the IHC of the different markers? I think it’ better to appreciate a co-localization. In addition, I would put figures 3 G and H immediately after figure 2D.
Round 2
Reviewer 1 Report
The authors answered to all my comments and the manuscript is now eligible for pubblication.
Reviewer 2 Report
The authors answered my concerns.